# Minimizing Polarization from Partially to Fully Observable Initial Opinions

## Abstract

This paper investigates the problem of minimizing polarization within a network, operating under the foundational assumption that the evolution of underlying opinions adheres to the most prevalent model, the Friedkin-Johnson (FJ) model. Although the objective function is non-convex, we show that for this problem, every local minimum is a global minimum. We extend this characterization to encompass a comprehensive class of matrix functions, including those pertinent to polarization and multiperiod polarization, even when addressing scenarios involving stubborn actors. Leveraging the geometry of the function, we propose a novel non-convex framework for this class of matrix functions and demonstrate its practical efficacy for minimizing polarization. Through empirical assessments conducted in real-world network scenarios, our proposed approach consistently outperforms existing state-of-the-art methodologies. Moreover, we extend our work to encompass a novel problem setting that has not been previously studied, wherein the observer possesses access solely to a subset of initial opinions. Within this agnostic framework, we introduce a non-convex relaxation methodology with similar theoretical guarantees to mitigate polarization.

## 1 Introduction

In recent times, there has been a notable surge in the utilization of social media, accompanied by its increasingly pivotal role in shaping the discourse of global politics. Prominent social networks such as Twitter, Mastodon, Reddit, and others have emerged as influential platforms for users to articulate their viewpoints and participate in socio-political dialogues. Ironically, the original intention of social media to foster connectivity among individuals has, at times, yielded an unintended consequence: the emergence of echo chambers. This phenomenon arises from the preferential attachment behavior exhibited by users who associate with others of similar inclinations, including shared political beliefs, as elucidated by Adamic & Glance (2005). Consequently, this trend has culminated in the polarization of active users within social media platforms along partisan lines, which, in turn, poses a potential threat to democratic ideals. The exposure of individuals primarily to like-minded peers reinforces their preexisting convictions, a phenomenon identified by Cass (2002). This reinforcement of congruent perspectives, in turn, steers users toward confirmation bias, inadvertently increasing the polarization of the network Kahneman (2011).

In today's society, minimizing polarization is crucial for fostering a sense of unity and constructive dialogue. By bridging divides and encouraging understanding, we can build a more resilient and inclusive community, enabling us to collectively address complex challenges and work towards shared goals. Polarization within social networking platforms can be attributed to a complex interplay between an individual's actions and the underlying social algorithms governing the provision of customized user experiences, encompassing features like personalized links and community recommendations Lazer (2015). Bakshy et al. (2015) delved into the impact of social media, exemplified by Facebook, on user perspectives and illuminated the salient role played by individual choices. These choices include interactions within one's social circles and the deliberate consumption of specific content, both of which wield substantial influence over the extent to which individuals are exposed to divergent ideological viewpoints. Consequently, comprehending polarization dynamics necessitates a profound understanding of the intricate processes through which people form their opinions and perspectives, rooted in the dual forces of social influence and selection.

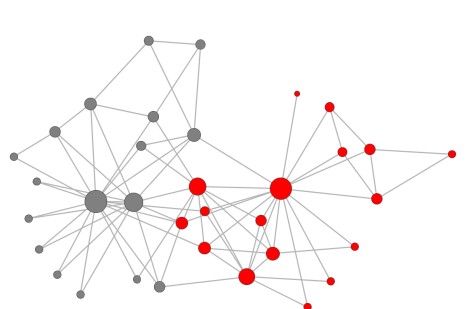
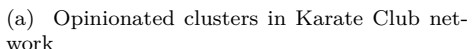

(a) Opinionated clusters in Karate Club network

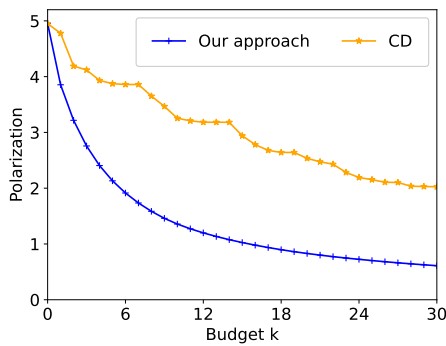

(b) Change in polarization using our proposed relaxation vs. the state of art approach (CD: Coordinate Descent) across varying budget on Karate Club network.

Figure 1: Reduction in Polarization on Karate Club Network

A vast amount of literature on opinion dynamics tries to model the evolution of opinions mathematically and study how it affects human behavior Bonabeau (2002); Centola (2018). Within the scope of this study, our primary emphasis centers on the examination of opinion dynamics as manifested within network structures. Among the well-recognized category of opinion dynamics models, a prominent subset is constituted by averaging models applied to networks. These models characterize an individual's opinion as a weighted aggregate of the opinions held by their neighbors in the network, a concept that has been extensively elaborated upon in Friedkin & Johnsen (1990); DeGroot (1974); Proskurnikov & Tempo (2017), and Abelson (1964). In this paper, we seek to understand how to strategically identify influential edges to minimize polarization while adhering to predefined budget constraints. For the rest of this paper, we assume that the underlying opinions evolve using one of the most popular averaging models, Friedkin and Johnsen's opinion formulation model, which incorporates the initial opinions of individuals into the averaging process.

**Motivation:** While many existing studies primarily center on reducing polarization by modifying individual opinions, our research takes a distinctive approach by emphasizing the utilization of network topology for this objective. This unique perspective provides guarantees of attaining a global minimum across the entire range of partially known to fully known initial opinions, an aspect that has been largely overlooked in prior research. To the best of our knowledge, we are the first to characterize and show global optimality results for these problems. Solving the optimization problem yields a Laplacian matrix whose structure, as explained in Section 6, elucidates the edges most influential in minimizing polarization. We aim to address the scenario outlined below.

*Instance*: Consider an undirected network denoted as $G$, characterized by $V$ users (nodes) and $E$ edges. Each user maintains an immutable initial opinion. The evolution of these opinions is governed by the Friedkin-Johnsen (FJ) opinion dynamics model. Within this framework, a budget denoted as $k$, where $k > 0$, can be allocated either for distribution among the existing edges of $G$ or for adding new edges to the network. Within this context, we pose the following research questions:

*Problem* 1. Given a graph and budget constraint $k$, how do we identify the optimal set of edges (together with edge weights) for minimizing polarization?

Figure 1 shows the reduction in polarization using our proposed non-convex relaxation on the classic Karate Club Network. This is described below.

While expressed or external opinions are empirically quantifiable, a fundamental limitation of the FJ model is the near impossibility of having prior knowledge of the initial opinions of all users. In many real-world scenarios, only a few users share their opinions about a topic on a social media platform, while many may prefer not to share their opinions publicly. In response to this challenge, we not only address the scenarios

where the user has complete knowledge of initial opinions but also expand our research to address an unexplored and novel problem setting where we have public access to only a subset of users' initial opinions.

*Problem* 2. Let $s$ represent the vector of initial opinions of users defined by $s = \begin{bmatrix} s_1 \\ s_2 \end{bmatrix}$, where $s_1$ denotes the vector containing the known initial opinions of users, and $s_2$ is the vector of the unknown initial opinions. How do we identify the optimal set of edges (together with edge weights) to minimize polarization while $s_2$ remains unknown?

The formal problem definitions are given after the introduction of the relevant literature and notation.

## 1.1 Main Contributions

In this subsection, we summarize the main contributions of our work.

**Global Optimality for both known and partially known initial opinions** We theoretically demonstrate that polarization under FJ dynamics can be minimized using simple tools such as gradient descent. We provide a general matrix result showing that every local minimum is a global minimum for a general class of matrix functions, $s^T M^{-k} s$, with $M \succ 0$, $s \in \mathbb{R}^n$ and an integer $k > 1$, where polarization and multiperiod controversy represent specific cases. [Theorem 4.2, Theorem 4.5]. We also extend this result to the presence of stubborn actors [Theorem 4.4]. We also provide a non-convex formulation with similar theoretical guarantees to minimize polarization when we have public access only to a partial set of users' initial opinions [Theorems 5.1 and 5.2]. Our proposed relaxations attain guarantees of a global minimum for minimizing polarization across all these scenarios. Utilizing projected gradient descent to solve these relaxations, we achieve notable improvements over existing state-of-the-art approaches, demonstrating superior performance even with fewer iterations [Section 6]. We also demonstrate empirically that our approaches are robust to small perturbations in estimating initial distributions. (Please see Appendix Section E.)

**A Novel Framework** Our contribution centers on providing the guarantees of global minimum for these non-convex functions together with a novel continuous optimization framework for minimizing polarization and multiperiod controversy, as well as polarization under stubborn actors. Instead of prescribing a particular method, we provide a general framework that can be employed with various randomized approaches and continuous optimization algorithms.

Our contributions provide a theoretical validation of the conjecture proposed in Chen et al. (2018), which states that the objective function for minimizing polarization, $s^T M^{-2} s$, where $M$ is a positive definite matrix, possesses a unique local minimum that is also the global minimum. We confirm that this function is free from non-global local minima and saddle points. This result is particularly useful for algorithm design, as it guarantees that first-order methods, such as projected gradient descent (PGD), will not become trapped at suboptimal points. While the prior works primarily focus on the polarization-disagreement index (Musco et al., 2018; Chen et al., 2018; Zhu et al., 2021), as a convex surrogate of the polarization function or study polarization under specific assumptions about the distribution of initial opinions (Chen et al., 2018), our work is the first to characterize a class of objective functions for which every local minimum is also a global minimum. We establish this result without relying on any assumptions regarding the distribution of opinions (as provided in Theorem 5.2).

## 1.2 Organization

The paper is structured as follows: Section 2 reviews the Friedkin-Johnsen model and the terminology pertinent to polarization. Section 3 discusses the prior related research. Section 4 is dedicated to a comprehensive theoretical examination of the objective function associated with polarization minimization. Section 5 provides non-convex formulations designed for scenarios where the observer has partial and complete access to users' initial opinions. Finally, Section 6 presents empirical findings relevant to the problem under investigation.

**Notation:** The set of natural and real numbers is denoted by $\mathbb{N}$ and $\mathbb{R}$, respectively. For a matrix $M$, $M_{ij}$ is the entry in the $i^{th}$ row and $j^{th}$ column. The identity matrix is represented as $I$. A vector of all ones is denoted by $\mathbf{1}$. The vectorized form of a matrix $M$ is denoted as $\text{vec}(M)$. The sets encompassing positive definite (PD) and positive semi-definite (PSD) matrices are respectively designated as $S_{++}^n$ and $S_+^n$. The Laplacian matrix of the adjacency matrix for graph G is denoted as $L$ and defined by the equation $L = D - W$, where $D$ is a diagonal matrix of (weighted) degrees associated with each node and $W$ is the weighted adjacency matrix. It is known that the graph Laplacian is a positive semi-definite matrix, and the set of Laplacian matrices $\mathcal{L}$ is a convex set. The algebraic connectivity of a given Laplacian matrix is provided by its second smallest eigenvalue, $\lambda_2$. We use Tr to denote the trace of the matrix. In the context of a vector $s$, $\|s\|_1$ and $\|s\|_2$ correspond to the $\ell_1$ and $\ell_2$ norms, respectively. Furthermore, the $\ell_0$ norm signifies the count of non-zero entries within the matrix or vector.

## 2 Preliminaries

In this section, we will review some of the most commonly used social influence models. We assume a real-valued, one-dimensional, continuous opinion space. In particular, we focus on linear continuous opinion models such as the DeGroot (1974) and Friedkin & Johnsen (1990). For simplicity, we choose the opinions to be scalar. Mathematically, they can also be a vector quantity representing an individual stance over various social phenomena.

### 2.1 French-DeGroot Model

French Jr (1956) proposed one of the first mathematical models for opinion formation and a group's collective behavior. Along these lines, DeGroot (1974) generalized this method and named it "iterative opinion pooling". This model describes a social learning process of opinion formation based on observing other individuals in the network. It formalizes when and how quickly several actors can reach a consensus of beliefs. In this model, the individuals' opinion is modeled as the harmonic average of the opinions of their neighbors in the network. Mathematically, the opinion update rule for estimates is given by the following equation:

$$z_i^{(t)} = \frac{1}{deg(i)} \sum_{j \in N(i)} w_{ij} z_j^{(t-1)} \ . \tag{1}$$

Here $w_{ij}$ represents the weight of $j$'s opinion on $i$, and the opinion of $i$ at time $t$ is written as $z_i^{(t)}$. The open neighborhood of vertex $i$ in $G$ is denoted by $N(i)$ and the degree of $i$ is represented by $deg(i)$. The DeGroot model always converges to consensus when the graph is connected.

### 2.2 Friedkin-Johnsen Model (FJ)

Friedkin and Johnsen generalized the DeGroot model by taking into account prejudice or initial opinions of individuals in the network Friedkin & Johnsen (1990). Let $s \in R^n$ represent the initial opinions of actors in the network. In the opinion dynamics process, this vector is assumed to be immutable. Let $z \in R^n$ denote the expressed opinions. Let $w_{ij} \geq 0$ denote the weight on edge $(i, j) \in E$. Fixed point iteration of the FJ opinion dynamics model is an extension to equation 1 and is given by

$$z_i^{(t)} = \frac{s_i + \sum_{j \in N(i)} w_{ij} z_j^{(t-1)}}{\sum_{j \in N(i)} w_{ij} + 1} \ . \tag{2}$$

At each time step, every actor adopts an expressed opinion that is proportional to the average of its own initial opinion and the opinion of its neighbors. It is well known that the equation 2 converge to an equilibrium set of opinions $z^*$ Bindel et al. (2015) given by

$$z^* = (I + L)^{-1} s \ . \tag{3}$$

In the above expression, $I$ is an Identity matrix, and $L$ is the combinatorial Laplacian of $G$ given by $D - W$. Note that $(I + L)$ is a positive definite matrix, and hence the inverse exists. From the equation (3), we can also observe that the expressed opinions are a contraction of initial opinions, i.e., $z_i$ is a convex combination of initial opinions of all nodes, including node $i$ in the network. Consensus is not guaranteed in FJ dynamics. Bindel et al. (2015) used this to quantify the price for not reaching the consensus. They show that updating $z_i$ as given in equation (2) is the same as minimizing the following quadratic function:

$$\min_{z_i} \quad (z_i - s_i)^2 + \sum_{j \in N(i)} w_{ij}(z_i - z_j)^2 \ .$$

The term $(z_i - s_i)^2$ is the stress incurred at node $i$ due to the difference between its initial and expressed opinions (also known as internal conflict) and the second term, $\sum_{j \in N(i)} w_{ij}(z_i - z_j)^2$, as the external conflict incurred due to the difference between the expressed opinions of the node $i$ and its neighbors.

## 2.3 In-Homogenous stubbornness in FJ model

The stubbornness of actors/nodes in the network is defined as the degree of resilience to change from their initial opinions. Recently, Xu et al. (2022) studied the Friedkin-Johnsen model in the presence of in-homogeneous stubbornness. The fixed point iteration of a node $i$ on a graph $G$ where every node has a certain degree of stubbornness to their initial opinions is then given as

$$z_i^{(t)} = \frac{k_i s_i + \sum_{j \in N(i)} w_{ij} z_j^{(t-1)}}{\sum_{j \in N(i)} w_{ij} + k_i} \ . \tag{4}$$

In the above equation, $k_i$ denotes the the degree of stubbornness and $k_i \geq 0$. By iterating the above equation, the expressed opinion vector at equilibrium $z^*$ is given as

$$z^* = (L + K)^{-1} K s \ , \tag{5}$$

where $K$ is a diagonal matrix with the degree of the stubbornness of each node in the network as its diagonal entries. From (5), we see that if the initial opinions of all nodes are perturbed by a constant $c$, the expressed opinions are changed to $z^* + c$.

## 2.4 Polarization under FJ dynamics

In this section, we formally define our problem and provide an array of definitions that are used in the literature. In the following, the notations $\bar{s}$ and $\bar{z}$ represent mean-centered initial opinions and expressed opinions, respectively. In the context of an undirected graph $G$ with associated initial opinions, $\bar{s}$, the expressed opinions at equilibrium are determined by the expression $\bar{z} = (I + L)^{-1} \bar{s}$ (Bindel et al., 2015).

**Definition 2.1** (Polarization). The polarization or controversy of an undirected network $G$ with Laplacian $L$ is defined as $\mathcal{P}(\bar{z}) = \bar{z}^T \bar{z} = \bar{s}^T (I + L)^{-2} \bar{s}$ (Chen et al., 2018; Musco et al., 2018).

Polarization formalizes how close the given network is to consensus, reflecting how far the steady-state opinions deviate from consensus. The polarization function is known to be non-convex (Rácz & Rigobon, 2023). We now formally describe the Problem 1 of minimizing polarization when the initial opinions are fully known:

**Minimizing Polarization for fully known Initial Opinions (Problem 1):** Given an undirected graph $G$ with adjacency matrix $A \in \{0,1\}^{n \times n}$ and its corresponding graph Laplacian $L_A$. Let $s$ denote the vector of initial opinions. Given a budget constraint $k \geq 0$, find an undirected graph $G'$ with adjacency matrix $A' \in \{0,1\}^{n \times n}$ and its corresponding graph Laplacian $L_{A'}$ that is at most $k$ edits (edge addition or removal) away from $G$ and minimizes the polarization. Formally, we solve:

$$\min_{L_{A'}} \quad \bar{s}^T (I + L_{A'})^{-2} \bar{s}$$
$$\text{subject to} \quad \| \operatorname{vec}(A) - \operatorname{vec}(A') \|_0 \leq 2k . \tag{6}$$

where $\| \operatorname{vec}(A) - \operatorname{vec}(A') \|_0$ represents the number of edge modifications with binary weights (additions or deletions) required to transform $G$ into $G'$. For the remainder of the paper, we omit subscripts when they are clear from context.

**Definition 2.2** (Disagreement). For a vector of expressed opinions, $\bar{z} \in \mathbb{R}^n$, the disagreement for a given undirected network $G$ with adjacency matrix $A$ is defined as

$$\mathcal{D}(\bar{z}) = \sum_{(i,j) \in E} A_{ij} (\bar{z}_i - \bar{z}_j)^2 .$$

The disagreement reflects the difference in the expressed opinion of a node with neighbors. The above definition can be expressed in matrix form using equation (3) as

$$\mathcal{D}(\bar{z}) = \bar{z}^T L \bar{z} = \bar{s}^T (I + L)^{-1} L (I + L)^{-1} \bar{s} .$$

**Definition 2.3** (Polarization-Disagreement Index). Polarization-Disagreement Index is defined as the sum of Polarization (Definition 2.1) and Disagreement (Definition 2.2) indices, given by $\mathcal{P}(z) + \mathcal{D}(\bar{z}) = \bar{s}^T (I + L)^{-1} \bar{s}$ (Chen et al., 2018; Musco et al., 2018).

The Polarization-Disagreement Index, as established in Musco et al. (2018), is a convex function and is commonly employed as a convex surrogate for the non-convex Polarization objective (Chen et al., 2018; Musco et al., 2018; Zhu et al., 2021). The following optimization function acts as a convex approximation to equation 6.

$$\min_{L_{A'}} \quad \bar{s}^T (I + L_{A'})^{-1} \bar{s}$$
$$\text{subject to} \quad \| \operatorname{vec}(A) - \operatorname{vec}(A') \|_0 \leq 2k . \tag{7}$$

**Average Conflict Risk.** The Average Conflict Risk (ACR) for polarization is defined by taking the expectation of all possible initial opinions. Akin to the setting in Chen et al. (2018), when the entries of the initial opinion vector $s$ are i.i.d. and sampled uniformly at random from $\{-1, 1\}^n$, such that $\mathbb{E}(ss^T) = I$, the ACR for polarization is defined as

$$ACR = E[s^T (I + L)^{-2} s] = E[Tr(s^T (I + L)^{-2} s)] = E[Tr(ss^T (I + L)^{-2})] = Tr((I + L)^{-2}) \tag{8}$$

In similar terms, the ACR for the polarization-disagreement index is given by $\operatorname{Tr}((I + L)^{-1})$ (Chen et al., 2018). Observe that $\operatorname{Tr}((I + L)^{-p})$, for $p \in \{1, 2\}$, is convex (proposition 10.6.17 from (Bernstein, 2009)). Thus, the Average Conflict Risk provides an alternative convex formulation to approximate the polarization function (equation 6) when the distribution of opinions is uniform. Note that the ACR formulation does not require the opinions to be mean-centered.

**Polarization in the presence of Stubborn Actors.** In opinion dynamics on graphs, polarization under stubbornness refers to the phenomenon where agents (nodes) with fixed or highly resistant opinions (referred to as stubborn agents) influence the equilibrium of the network, preventing full consensus and leading to persistent disagreement across the network. Stubbornness induces higher polarization, as stubborn nodes anchor parts of the network to differing opinion values, preventing full convergence. Xu et al. (2022) defined polarization with stubborn actors in the Friedkin-Johnsen model as follows:

**Definition 2.4** (Polarization under stubbornness). Given an undirected network, $G$ with initial opinions, $s$, expressed opinions $z$, and the stubbornness matrix $K$ denoting the degree of stubbornness, the polarization with stubbornness is defined as $\mathcal{P}(z) = \sum_{i \in V} k_i z_i^2 = \bar{s}^T K (L + K)^{-1} K (L + K)^{-1} K \bar{s}$, where $k_i$ denotes the degree of stubbornness of node $i$.

When $K = I$, this definition reduces to non-mean-centered polarization of expressed opinions. Xu et al. (2022) provided a different notion of mean-centeredness for polarization in the presence of stubborn actors. If $\mathbf{1}^T K s \neq 0$, then $s$ is changed to $\bar{s} = s - \frac{\mathbf{1}^T K s}{n} \mathbf{1}$ and consequently the expressed opinions $z$ is changed to $\bar{z} = z - \frac{\mathbf{1}^T K s}{n} \mathbf{1}$. We use $s$ and $z$ instead of $\bar{s}$ and $\bar{z}$ for consistent notation in the theoretical results pertinent to stubborn actors.

**Multiperiod Setting.** So far, we have considered a single time period polarization. As an extension, it is natural to consider a similar objective over a prolonged time instance. We consider a $\mathcal{T}$-period controversy as an extension to one-period polarization defined in Definition 2.1. In the first time period, the expressed opinions $z(\mathcal{T}(1))$ are $(I + L)^{-1}s$. These become the initial opinions for the next subsequent step, and the expressed options at the second period become $z(\mathcal{T}(2)) = (I + L)^{-2}s$. The polarization of these opinions is then added to the initial polarization. This process is repeated for $\mathcal{T} + 1$ time steps, where $\mathcal{T} \in \mathbb{N} \cup \{\infty\}$. This scenario is formulated as controversy but not polarization as after each time period, $z(\mathcal{T}(i))$ need not be mean-centered Musco et al. (2018); Chen et al. (2018). In a multi-period setup, the objective is to minimize controversy across all time periods. By incorporating this, we get the following framework:

$$\min_{L \in \mathcal{L}} s^T [(I + L)^{-2} + (I + L)^{-4} + \cdots (I + L)^{-2\mathcal{T}-2}]s \ . \tag{9}$$

## 3 Prior work

Numerous researchers across the scientific community have been actively engaged in the study of polarization and its associated characteristics. Previous research on polarization minimization can be broadly classified into two categories: one approach centers on diminishing polarization by introducing perturbations to initial opinions, while the other attains polarization reduction through modifications to the network structure. In this work, our primary focus lies in the domain of reducing polarization by altering the network structure. For a broader review of other related research pertinent to the first category, please see Appendix A.

We first discuss the related work pertinent to Problem 1. Musco et al. (2018) delved into the problem of determining an undirected graph topology with a prescribed edge count to minimize polarization and disagreement. Their work established the convexity of the network's Polarization-Disagreement (PD) index with respect to the Laplacian matrix $L$. Moreover, they provided proof of the existence of a graph topology with $\mathcal{O}(\frac{n}{\epsilon^2})$ edges, approximating the optimum within a factor of $(1+\epsilon)$ through the utilization of Spielman and Srivastava's sparsification algorithm based on effective resistance (Spielman & Srivastava, 2008). Chen et al. (2018) defined polarization as the sum of squares of expressed opinions and proposed a measure called ACR (equation 8) to minimize polarization in the presence of an unknown opinion vector. Chitra & Musco (2020) augmented the Friedkin-Johnsen (FJ) model by establishing connections between users who share matching ideologies, aiming to minimize disagreement among users. On similar lines, Gaitonde et al. (2020) showed that the entire graph spectra of the Laplacian matrix are relevant rather than their extreme eigenvalues to maximize repeated disagreement in a network. Neumann et al. (2024) showed that polarization and related measures could be approximated in sublinear time when the initial opinions are not known.Bhalla et al. (2023a) extended the FJ model and showed how polarization increases via swaps of more agreeable opinionated edges for more disagreeable ones. Recently, Rácz & Rigobon (2023) studied how an administrator or a centralized planner can alter the network to reduce polarization. They show the nonconvexity of the polarization function and bound its value using the Cheeger constant Chung (1997). Furthermore, they show that the value of polarization is not monotonic by the addition of edges unless the initial opinions vector is chosen to be the eigenvector corresponding to the second smallest eigenvalue of $L$. Rácz & Rigobon (2023) explored the Fiedler difference vector approach (FD) and the coordinate descent approach (CD) as mechanisms for polarization reduction and observed that FD effectively reduces polarization without diminishing network homophily, which is defined as a tendency where similar individuals connect to each other. In the CD approach, non-edges that yield the most significant polarization reduction are iteratively added to the graph until the budget constraint is satisfied. We employ CD, FD, and ACR (defined in 8) approaches as baselines for comparative evaluation against our proposed non-convex relaxations in Section 6.

Since Problem 2 has never been dealt with before, no prior work is dedicated to it. However, related research exists in the limiting case where none of the initial opinions are observed, effectively reducing it to the problem of ACR (8) Chen et al. (2018) (Further research pertinent to FJ dynamics is provided in Appendix A).

## 4 Theoretical Results

In this section, we study the global optimality of polarization. To that end, we show that it falls under a special kind of non-convex function, namely the invex function. Invex functions can be seen as a generalization of convex functions. Hanson (1981) defined invexity as follows.

**Definition 4.1.** Let $f(\theta)$ be a function defined on a set $\mathcal{C}$. Let $\eta$ be a vector-valued function defined in $\mathcal{C} \times \mathcal{C}$ such that the Frobenius inner product, $\langle \eta(\theta_1, \theta_2), \nabla f(\theta_2) \rangle$, is well defined $\forall \theta_1, \theta_2 \in \mathcal{C}$. Then $f(\theta)$ is a $\eta$-invex function if $f(\theta_1) - f(\theta_2) \geq \langle \eta(\theta_1, \theta_2), \nabla f(\theta_2) \rangle, \forall \theta_1, \theta_2 \in \mathcal{C}$.

A function is an invex function iff it attains global minima at every stationary point Ben-Israel & Mond (1986). Next, we prove the invexity of a general class of functions. While this result can be of independent interest, we restrict our attention to minimizing polarization and related problems. By little abuse of notation, we represent $\eta$ as a vector or matrix, depending on the specific context, in order to enhance the clarity of our presentation when the implications of such a representation are readily discernible.

**Note**: All the proofs are in the supplementary material.

**Theorem 4.2.** *The class of matrix functions $f(M) = s^T M^{-k} s$, with $M \succ 0$ and any integer $k > 1$ are $\eta$-invex for $\eta(\cdot, M) = M$.*

**Corollary 4.3.** *As a consequence of Theorem 4.2, the polarization function, $f(L) = s^T (I+L)^{-2} s$, is $\eta$-invex for $\eta(\cdot, L) = I + L$.*

From the above corollary, we deduce that every local minimum of the polarization objective function is a global minimum. The nonconvexity of the function $s^T M^{-2} s$ for $M \succ 0$ can be shown by restricting it to a line. For example, plot of $f(z) = s^T \begin{bmatrix} z & 0.9 \\ 0.9 & 1 \end{bmatrix}^{-2} s$ with respect to $z \in [1, 2]$ and $s = \begin{bmatrix} 1 \\ 1 \end{bmatrix}$ is visibly non-convex (the figure is provided in the supplementary material section C). Thus, $s^T M^{-2} s$ is a non-convex but invex function. In the following Theorem, we show that the polarization remains invex even in the presence of stubborn actors.

**Theorem 4.4.** *Let $K$ represent the diagonal matrix of stubbornness coefficients associated with stubborn actors in the network. The polarization function $f(L) = s^T K (L + K)^{-1} K (L + K)^{-1} K s$ is $\eta$-invex for $\eta(\cdot, L) = \frac{(L+K)}{2}$.*

Thus, even with the presence of stubborn actors, every local minimum is also a global minimum for the function $s^T K (L + K)^{-1} K (L + K)^{-1} K s$ under FJ dynamics. This remains true even in the multi-period setting described below.

**Theorem 4.5.** *The multiperiod controversy, i.e., the objective function given in equation 9, is $\eta$-invex for $\eta(\cdot, L) = I + L$.*

The following Proposition quantitatively characterizes the global minimum and helps us understand the graph structures where the global minimum is attained for multiperiod polarization.

**Proposition 4.6.** *The global minimum for multiperiod polarization is attained for complete graphs.*

**Theorem 4.7.** *Let $G$ be an undirected graph with its associated Adjacency matrix $A \in \{0, 1\}^{n \times n}$ and its graph Laplacian $L$. Let the budget $k$ denote the number of graph edits in terms of addition of edges. For a choice of initial opinions vector, identifying a graph Laplacian, $L$, nearest to the given graph Laplacian, $L_0$, within $k$ edits and having minimum polarization is $\mathcal{NP}$-hard.*

The proof relies on showing the equivalence between the following two optimization problems.

$$\arg\min_{L \in \mathcal{L}} \max_{s \in R^n, s \perp 1, \|s\|_2^2 \le 1} \quad s^T (I + L)^{-2} s$$
$$\text{subject to} \quad L_{ij} \in \{-1, 0\}, \text{ for } i \ne j$$
$$\| \operatorname{vec}(L) - \operatorname{vec}(L_0) \|_0 \le 4k \ ,$$

and

$$\arg\max_{L \in \mathcal{L}} \quad \lambda_2(L)$$
$$\text{subject to} \quad L_{ij} \in \{-1, 0\}, \text{ for } i \ne j$$
$$\| \operatorname{vec}(L) - \operatorname{vec}(L_0) \|_0 \le 4k \ .$$

The theoretical results established in Theorems 4.2, 4.4, and 4.5, along with Corollary 4.3, collectively imply that every local minimum is also a global minimum for functions of the form $s^T M^{-k} s$, where $M \succ 0$. Typically, it is not easy to minimize such an objective function with a budget constraint. In fact, Theorem 4.7 presents an $\mathcal{NP}$-hardness result for a restricted variant of the problem. Although, we also note that it does not necessarily imply that the primary formulation with a fixed vector $s$ presented in equation 6 is also $\mathcal{NP}$-hard. This restricted setting motivates the consideration of a continuous relaxation, specifically an $\ell_1$-based approach, for minimizing polarization. The computational complexity of the original problem of minimizing polarization, as described in equation 6, remains an open question.

## 5 Nonconvex relaxation for minimizing polarization

While Theorem 4.2 and Theorems 4.5 establish that polarization and multiperiod polarization are invex functions, they do not readily provide a framework to solve them. Next, we develop a non-convex relaxation framework for Problem 1 and 2 to minimize polarization. We first delve into a scenario where the observer is limited to accessing only a subset of the users' initial opinions within the network (Problem 2). The vector of initial opinions of users, denoted as $s = \begin{bmatrix} s_1^T & s_2^T \end{bmatrix}^T$, is partitioned into two components: $s_1$, comprising the known initial opinions of users, and $s_2$, representing the initial opinions that remain concealed from the observer. We assume that $s_2$ follows a distribution characterized by a zero mean and an identity covariance matrix, such as the standard Gaussian or uniform distributions. Formally, we take $\mathbb{E}(s_2) = 0$ and $\mathbb{E}(s_2 s_2^T) = I$ (we relax the latter assumption in Theorem 5.2). Let us represent $(I + L)^{-2}$ as $\begin{bmatrix} W_{11} & W_{12} \\ W_{12} & W_{22} \end{bmatrix}$, with each $W_{ij}$ being a block matrix having appropriate dimensions. For the sake of clarity, we omit the dimension details when they are evident from the context. Using the definition of polarization, we obtain:

$$f(L) = s^T (I + L)^{-2} s = \begin{bmatrix} s_1^T & s_2^T \end{bmatrix} \begin{bmatrix} W_{11} & W_{12} \\ W_{12} & W_{22} \end{bmatrix} \begin{bmatrix} s_1 \\ s_2 \end{bmatrix}$$
$$= s_1^T W_{11} s_1 + s_1^T W_{12} s_2 + s_2^T W_{12} s_1 + s_2^T W_{22} s_2$$

It is important to highlight that $f(L)$ is a random variable due to $s_2$. Therefore, our objective is to minimize the expected polarization. Taking the expectation on both sides leads to the following:

$$\mathbb{E}(f(L)) = \mathbb{E}(s_1^T W_{11} s_1) + \mathbb{E}(\operatorname{Tr}(W_{22} s_2 s_2^T)) \tag{10}$$

While a two-step approach involving the initial minimization of $s_1^T W_{11} s_1$ followed by the minimization of $\operatorname{Tr}(W_{22})$ might seem appealing, the budget constraint prohibits their decoupling. Our subsequent result establishes that the expected polarization $\mathbb{E}(f(L))$ is an invex function. We now proceed to formally define Problem 2 as a constrained minimization problem.

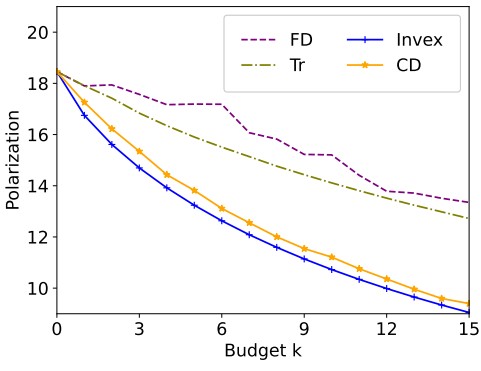 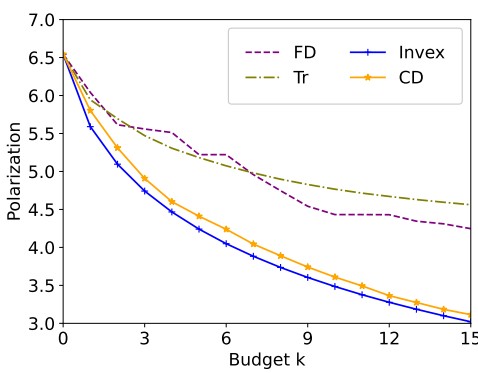

(a) Change in polarization with budget k in SBM when an initial opinion of "-1" is assigned to one community of nodes and an opinion "+1" to the other community.

(b) Change in polarization for uniformly distributed opinions within each community in SBM.

Figure 2: Reduction in Polarization on Stochastic Block Model

**Minimizing the Expected Polarization for Partially known Initial Opinions (Problem 2):** For a given adjacency matrix $A$, let $L_A$ denote the corresponding graph Laplacian. Within the setting described in Section 5, our objective is to construct an adjacency matrix $A'$ by making at most $k$ edits to the given adjacency matrix $A$ such that $\mathbb{E}(f(L))$ is minimized. Formally:

$$\min_{A'} \quad \mathbb{E}(s_1^T W_{11} s_1) + \mathbb{E}(\text{Tr}(W_{22} s_2 s_2^T)) \tag{11}$$
$$\text{subject to} \quad \|\text{vec}(A) - \text{vec}(A')\|_0 \leq 2k \,,$$

where $W_{11}$ and $W_{22}$ are matrix elements from the block-matrix decomposition of $(I + L_{A'})^{-2}$.

We now proceed to characterize the objective function of equation 11.

**Theorem 5.1.** *Given a vector $s \in \mathbb{R}^n$ defined as $s = \begin{bmatrix} s_1^T & s_2^T \end{bmatrix}^T$, where $s_1 \in \mathbb{R}^{n-m}$ and $s_2 \in \mathbb{R}^m$, and assuming that $s_2$ is selected from a distribution satisfying $\mathbb{E}(s_2) = 0$ and $\mathbb{E}(s_2 s_2^T) = I$, it follows that $\mathbb{E}(f(L))$ is invex.*

This result stems from the observation that the expected polarization can be expressed as a summation of invex functions. To illustrate this, we rephrase the expected polarization as $\mathbb{E}(f(L)) = a^T (I + L)^{-2} a + \sum_{i=1}^m b_i^T (I + L)^{-2} b_i$, where $a = \begin{bmatrix} s_1^T & 0 \end{bmatrix}^T$ and $b_i = \begin{bmatrix} 0 & e_i^T \end{bmatrix}^T$ for all $i = \{1, \cdots, m\}$, with $e_i \in \mathbb{R}^m$ denoting the standard unit vector containing a 1 at its $i$-th entry. We propose the following continuous relaxation ($\ell_1$) for this scenario:

$$\min_{L} \quad a^T (I + L)^{-2} a + \sum_{i=1}^m b_i^T (I + L)^{-2} b_i \tag{12}$$
$$\text{subject to} \quad L \in \mathcal{L}$$
$$\|\text{vec}(L) - \text{vec}(L_0)\|_1 \leq 4k \,.$$

The $\ell_1$ relaxation reformulates the original combinatorial problem by replacing the constraint of at most $k$ unit-weight edge modifications (additions or deletions) with an $\ell_1$-norm constraint, allowing the budget $k$ to be distributed as fractional weights over the edge set.

The following Theorem generalizes Theorem 5.1, with less restrictive assumptions concerning the distribution of the unknown initial opinions $s_2$. While we maintain the assumption of zero mean for these opinions, we now allow for a more general covariance matrix.

**Theorem 5.2.** *Given a vector $s \in \mathbb{R}^n$ defined as $s = \begin{bmatrix} s_1^T & s_2^T \end{bmatrix}^T$, where $s_1 \in \mathbb{R}^{n-m}$ and $s_2 \in \mathbb{R}^m$, and assuming that $s_2$ is selected from a distribution satisfying $\mathbb{E}(s_2) = 0$ and $\mathbb{E}(s_2 s_2^T) = \Sigma$, it follows that $\mathbb{E}(f(L))$ is invex.*

It is worth noting that the proposed non-convex (Invex) formulation framework provides a generalization of the established Average Conflict Risk (ACR) measure (8) for the purpose of polarization minimization. Observe that we relax the non-convex budget constraint $\ell_0$ to $\ell_1$ and express it in terms of Laplacian rather than adjacency matrix (unlike stated in equation (6)). The budget constraint has been modified to $4k$ instead of $2k$ because it affects *four* entries of the Laplacian matrix ($\{(i,j), (j,i), (i,i), (j,j)\}$).

When all initial opinions are known (Problem 1), i.e., $s = s_1$, optimization problem 12 simplifies to:

$$
\begin{aligned}
\min_{L} \quad & s^T (I + L)^{-2} s \\
\text{subject to} \quad & L \in \mathcal{L} \\
& \| \operatorname{vec}(L) - \operatorname{vec}(L_0) \|_1 \leq 4k \ .
\end{aligned}
\tag{13}
$$

This is a result of $\sum_{i=1}^{m} b_i^T (I + L)^{-2} b_i = 0$ as the second term from the optimization problem 12 vanishes when all the opinions are known. In this paper, we aim to solve the optimization problems 12 and 13. A practical limitation when solving such non-convex formulations is that the resulting Laplacian can become dense. Even for smaller budgets, we observed that the solution tends to converge to a complete graph with smaller weights distributed across the network. To address this, we further prune the solution obtained by using a thresholding parameter $\rho$ to discard smaller weights in $L$ and set them to zero. Notice that after pruning the resultant matrix, $\hat{L}$ need not be a Laplacian. We get the optimal Laplacian $L^{proj}$ closest to $\hat{L}$ by projecting the diagonal entries Sato (2019):

$$
L_{ii}^{proj} = - \sum_{j=1, j \neq i}^{n} \hat{L}_{ij}, \forall i \in \{1, \cdots, n\}
$$

Only the diagonal entries need to be updated after pruning. The non-convex relaxations mentioned above can be readily extended to address multiperiod polarization and polarization scenarios involving stubborn actors due to the invex nature of the objective functions (Theorem 4.4 and 4.5). It is worth noting that any first-order algorithm should be applicable to our framework to attain global optimality. We use the projected gradient descent (PGD) algorithm to solve the optimization problems 12, 13. In the next section, we empirically demonstrate that our relaxations lead to better minima with a few iterations of PGD.

## 6 Experimental Results

In this section, we demonstrate the effectiveness of our method in mitigating polarization across diverse networks.

**Multi-period Scenario** : Note that the Laplacian that minimizes single-period polarization also minimizes multi-period polarization. In this section, we provide experimental details on single-period polarization (optimization problems 12 and 13) and the performance of various approaches to minimize multi-period controversy (equation 9) under budget constraints can directly be inferred from their performance in minimizing single-period polarization.

### 6.1 For known initial opinions (Problem 1)

Apart from the Coordinate Descent approach (CD) proposed by Rácz & Rigobon (2023), two other approaches to minimize polarization are to minimize $\operatorname{Tr}((I + L)^{-2})$ (ACR: equation 8) and maximize $\lambda_2(L)$ Ghosh & Boyd (2006); Wang & Van Mieghem (2010). The heuristic approach to maximize $\lambda_2(L)$ is based on adding edges between nonadjacent vertices in the graph that have the largest absolute difference in the entries of Fiedler vector Chung (1997). In this section, we compare the empirical performance of our non-convex

(invex) relaxation (equation 13) with the Coordinate Descent approach (CD) proposed by Rácz & Rigobon (2023), ACR (Trace minimization) and Fiedler Difference vector (FD) Wang & Van Mieghem (2010). We use the projected gradient descent method (PGD) in CVX Diamond & Boyd (2016); Agrawal et al. (2018) to solve our proposed non-convex relaxation. We study the performance of our approach on real-world and synthetic networks. For synthetic networks, we consider the stochastic block models. Additional analysis on the sensitivity of proposed methods to perturbations in initial opinions is given in Appendix E.

**Stochastic Block Model:**   The Stochastic Block Model (SBM) generates random graphs with inherent community structure, emphasizing node groups. In our simulation, we create two communities, each with 250 nodes. Inter-cluster and intra-cluster densities are 0.02 and 0.08, resulting in 500 nodes and 6,359 edges in the network. We distribute initial opinions in two ways: (1) assigning "-1" to one block and "+1" to the other, creating well-connected opinionated clusters (see Figure 2(a)), and (2) uniformly distributing "+1" and "-1" opinions within each block (Figure 2(b)). Across both scenarios, the invex relaxation method consistently outperforms the Coordinate Descent, Tr, and FD methods. We use the thresholding parameter $|\rho| = 0.0002$, step size $\alpha = 0.5$, and run PGD for 100 iterations. In the first scenario, with distinctly separated opinionated clusters, the average number of edges using our proposed non-convex (invex) relaxation with thresholding parameter $\rho$ is 7,942. In the second scenario, with uniform opinion distribution, it is 7,616 (after thresholding).

Our empirical analysis shows that our proposed non-convex relaxation consistently outperforms other methods in reducing polarization. The Fiedler Difference (FD) approach primarily aims to reduce polarization by increasing algebraic connectivity. While raising the second smallest eigenvalue ($\lambda_2$) may cause other eigenvalues to increase as $L \in S_+^n$, this increase is insufficient for FD to achieve significant polarization reduction. In the second scenario of our construction of SBM, the FD approach seeks to maximize $\lambda_2$ by introducing additional edges within the opinionated clusters, potentially inadvertently fostering the creation of echo chambers.

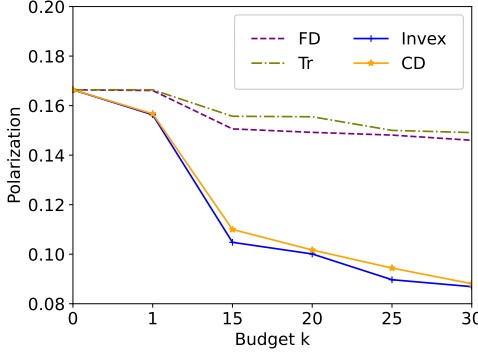 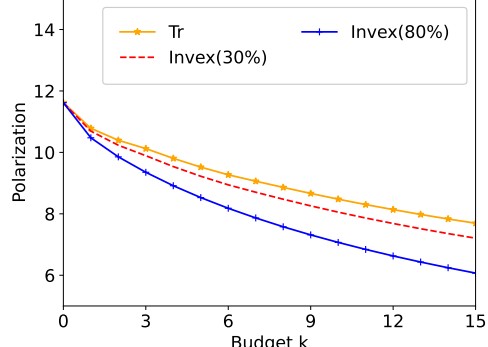

(a) Change in polarization using our proposed non-convex relaxation, CD, Trace and FD on the Twitter network.

(b) Change in polarization with budget for partially observable opinions of (30% and 80% of known initial opinions) using invex relaxation and $\mathrm{Tr}((I + L)^{-2})$

Figure 3: Reduction in Polarization on the Twitter network (Problem 1) and on SBM with partially observable initial opinions (Problem 2)

**Twitter:**   The Twitter dataset, originally gathered for the analysis of the Delhi legislative assembly elections debate by De et al. (2014) through hashtags such as #BJP, #AAP, #Congress, and #Polls2013, comprises an undirected network involving 548 users with a total of 3638 interactions. Initial opinions are derived from user interactions on Twitter employing sentiment analysis. Figure 3(a) illustrates the polarization variation across different budgets ($k = 1, 15, 20, 25, 30$) using our non-convex relaxation (optimization problem 13), CD, Trace minimization, and FD methodologies. The projected gradient descent method for optimization problem 13 is executed for a maximum of 130 iterations across all budgets, with a step size of $\alpha = 0.5$ and

a thresholding parameter $|\rho| = 0.0002$. Notably, the reduction in polarization is most pronounced when employing our non-convex relaxation.

**The US Senate:** This network captures the co-sponsorship of bills among US senators during session 114, as documented by Neal (2022). In this representation, each senator assumes the role of either a sponsor or co-sponsor of a bill, and edges between senators signify their joint co-sponsorship of a bill during that session. Recent studies, such as those by Hohmann et al. (2023) and Neal (2020), have explored the relevance of such co-sponsorship networks in the context of polarization. This particular network encompasses a total of 102 nodes, with 46 Democrats, 54 Republicans, and 2 Independents, interconnected by 1832 edges. We assign an initial opinion of "+1" to Democrats, "−1" to Republicans, and "0" to Independents.

Figure 4 visually presents the polarization reduction achieved using our proposed invex relaxation (optimization problem 13), comparing it to the Coordinate Descent Rácz & Rigobon (2023), the Tr minimization, and the Fiedler Difference (FD) approaches. In our computational experiments, we ran projected gradient descent for 100 iterations, employing a step size of $\alpha = 0.2$ and setting $|\rho| = 0.0002$. The average number of edges added across all budgets amounts to 2436. The results, as depicted, demonstrate that our invex relaxation significantly outperforms all existing approaches in terms of minimizing polarization.

**Polbooks:** This network comprises books related to US politics and was compiled during the 2004 presidential election, as documented by Rossi & Ahmed (2015). The network includes 105 users with 441 interactions. Interactions within the network reflect instances where customers on the Amazon platform frequently purchased these books together. The books are categorized based on their political leanings, falling into three categories: Liberal, Conservative, or Neutral. Specifically, there are a total of 43 books classified as Liberal, 49 as Conservative, and 13 as Neutral. We assign an initial opinion of " + 1" to Liberal, " − 1" to Conservative, and "0" to Neutral. Figure 4(a) illustrates the variation in polarization across different budgets. The projected gradient descent for invex relaxation is executed for a maximum of 100 iterations, utilizing a step size of $\alpha = 0.2$ and $|\rho| = 0.0002$.

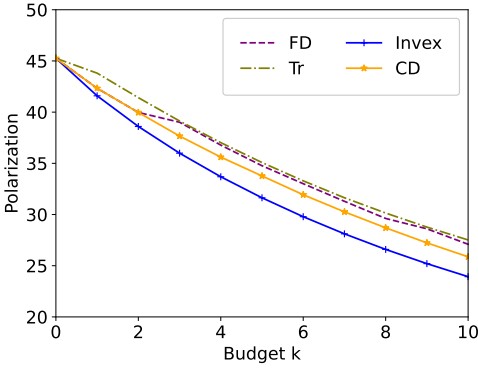

(a) Change in polarization with budget using invex relaxation, CD, Tr and FD on Polbooks

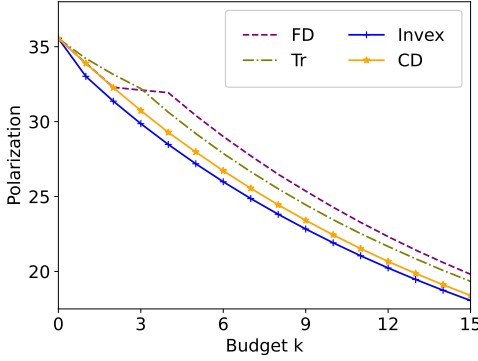

(b) Reduction in polarization with budget using invex relaxation, CD, Tr, and FD approaches on the US Senate Network.

Figure 4: Reduction in Polarization on Polbooks and US Senate networks

## 6.2 For Partially observable initial opinions (Problem 2)

In this section, we study the empirical performance of our proposed invex relaxation method, as presented in equation 12, and the ACR measure defined in 8. It's worth noting that equation 12 serves as a generalization of the ACR measure.

**Stochastic Block Model:** We generate an SBM model using the parameters as described in 6.1, where the unknown initial opinions of users are drawn from a uniform distribution over all vectors in $\{-1, +1\}^n$. Figure 3(b) illustrates the polarization variation with the budget, considering scenarios where the observer

possesses access to 30% and 80% of users' initial opinions. We experimented on two partial observable percentages of initial opinion. It is evident that our proposed non-convex (invex) relaxation consistently outperforms the Average Conflict Risk (ACR) measure and is equal to its value $\text{Tr}(I + L)^{-2}$ only when the observer has no knowledge of any users' opinions.

To facilitate our experimentation with Coordinate Descent, we estimate unknown opinions using mean imputation, specifically setting $s_2 = \text{mean}(s_1)$. The corresponding outcome is illustrated in Figure 5. It is evident that CD outperforms Trace when it has access to a larger percentage of initial opinions.

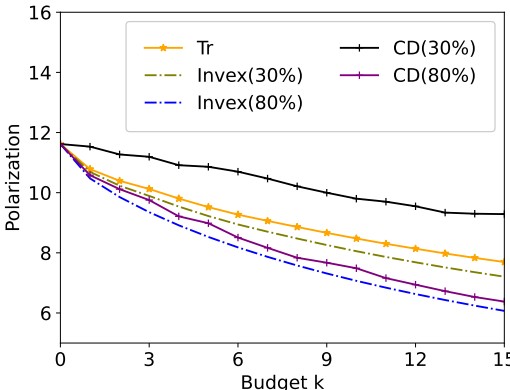

Figure 5: Change in polarization with budget for partially observable opinions of (30% and 80% of known initial opinions) using invex relaxation, CD (with mean imputation, i.e., $s_2 = \text{mean}(s_1)$) and $\text{Tr}((I + L)^{-2})$

**Interpretation in social context:** Based on empirical observations, our optimization approaches presented in 12 and 13 effectively minimize polarization by introducing additional edges among users with polarized opinions. This aligns with findings from previous research, including Wang & Kleinberg (2023); Chitra & Musco (2020); Rácz & Rigobon (2023). Utilizing continuous relaxation techniques as demonstrated in the optimization problems 12 and 13, we can identify significant interactions within a social network, typically represented by edges with high weights that play a pivotal role in the minimization of polarization. Armed with this insight, social algorithms can offer link recommendations and promote exposure to diverse content among network users. This strategic approach helps prevent the reinforcement of like-minded opinions, ultimately contributing to the reduction of polarization within the network.

## 7 Conclusion and Future Directions

This paper addresses polarization mitigation by altering network topology in two scenarios: when initial opinions are known and when the observer has partial knowledge of the opinions. We introduce a novel non-convex relaxation framework for known opinions and demonstrate the projected gradient descent's efficacy in polarization minimization. We extend this to scenarios with incomplete knowledge of initial opinions, proposing a novel non-convex formulation that generalizes the ACR (trace minimization) approach. Continuous relaxation techniques, as shown in 12 and 13, identify pivotal interactions that can be leveraged to provide link recommendations and diversify content exposure to mitigate polarization. Existing scalability studies primarily focus on the computation of the polarization, denoted as $s^T(I + L)^{-2}s$ Xu et al. (2021). In the future, it might be of significant interest to explore the applicability of randomized algorithms in conjunction with our findings to minimize polarization for larger network configurations.

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

## A Further Related work

In this section, we present additional references that pertain to research on polarization using FJ dynamics. Guerra et al. (2013) identified a characteristic of polarized networks, namely, a lower concentration of high-degree nodes in the vicinity of boundaries separating distinct communities. Zhu et al. (2021) provided a scalable greedy algorithm for optimizing the polarization-disagreement index for a given graph by adding a set of edges. They show that the index is monotonic with respect to the addition of edges, and despite the function being non-submodular, they provided a bounded approximation ratio. Chen & Rácz (2021) explored the amplification of disagreement and polarization through perturbations in the initial opinions held by network nodes. Bhalla et al. (2021) scrutinized the dependence of polarization on localized edge dynamics, revealing that the introduction of an edge between closely affiliated like-minded users leads to an increase in polarization. Matakos et al. (2017) provided greedy heuristics to minimize polarization by perturbing initial opinions. Recently Zhu & Zhang (2022) provided a linear time approximation algorithm for minimizing the risk of conflict on social networks. Xu & Zhang (2023) propose a greedy algorithm for optimizing the effective resistance between a set of leader nodes and the rest of the nodes in a noisy social network. The proposed algorithm adds new edges to the network, with each edge incident to a node in the leader group, in order to minimize the effective resistance and thus reduce polarization in the leader-follower opinion dynamics. Bhalla et al. (2023b) find that the combined presence of confirmation bias and friend-of-friend link recommendations result in high polarization in a time-evolving network with a variant of the Friedkin-Johnsen opinion dynamics. Wang & Kleinberg (2023) study the relationship between relevance and conflict in social networks in the context of link recommendations. Using Friedkin-Johnsen as the opinion dynamics model, they derive a closed-form expression to study the amount of change in opinion conflict caused by general link additions and show that link addition does not increase opinion conflict. They also introduce a measure to empirically evaluate a recommendation algorithm's ability to reduce conflict and show that some of the recommendation algorithms that are more accurate on real-world social networks effectively reduce conflict.

## B Proofs of theorems and lemmas

### B.1 Proof of Theorem 4.2

*Proof.* We first compute the gradient of the function. Let $X = M^k$. Then $f(M) = s^T (X)^{-1} s$. It is known that:

$$\frac{\partial X^{-1}}{\partial m_{ij}} = -X^{-1} \frac{\partial X}{\partial m_{ij}} X^{-1} . \tag{14}$$

By product rule,

$$\frac{\partial M^k}{\partial m_{ij}} = J^{ij} M^{k-1} + M J^{ij} M^{k-2} + \cdots + M^{k-1} J^{ij} , \tag{15}$$

where $J^{ij}$ is the matrix with 1 at $(i,j)^{th}$ entry and zero else where. By substituting equation 15 in equation 14, we get:

$$\frac{\partial s^T X^{-1} s}{\partial m_{ij}} = -s^T M^{-k} J^{ij} M^{-1} s - s^T M^{-(k-1)} J^{ij} M^{-2} s - \cdots - s^T M^{-1} J^{ij} M^{-k} s .$$

Considering $M^{-l}$ as $A$ and $M^{-q}$ as $B$ and using identity that $s^T A J^{ij} B s = (A^T s s^T B^T)_{ij}$ (equation 454 from Petersen et al. (2008)) we get

$$\frac{\partial s^T X^{-1} s}{\partial m_{ij}} = -(M^{-k} s s^T M^{-1})_{ij} - (M^{-(k-1)} s s^T M^{-2})_{ij} - \cdots - (M^{-1} s s^T M^{-k})_{ij} .$$

This implies:

$$\frac{\partial s^T M^{-k} s}{\partial M} = -M^{-k} s s^T M^{-1} - M^{-(k-1)} s s^T M^{-2} - \cdots - M^{-1} s s^T M^{-k} . \tag{16}$$

equation 16 represents the gradient of the function $s^T M^{-k} s$ with respect to $M$. Let $M, N \in S_{++}^n$. To show invexity for function a $f$, we need to show that there exists an $\eta(N, M)$ such that

$$f(N) - f(M) \geq \langle \eta(N, M), \nabla f(M) \rangle .$$

In our case, this implies that we need to show the existence of $\eta(N, M)$ such that

$$s^T N^{-k} s - s^T M^{-k} s \geq \left\langle \eta(N, M), \frac{\partial s^T M^{-k} s}{\partial M} \right\rangle .$$

After substituting for the gradient, we get

$$s^T N^{-k} s - s^T M^{-k} s \geq - \left\langle \eta(N, M), M^{-k} s s^T M^{-1} \right\rangle - \cdots - \left\langle \eta(N, M), M^{-1} s s^T M^{-k} \right\rangle .$$

With little algebraic manipulation, we can write

$$s^T N^{-k} s - s^T M^{-k} s \geq - \operatorname{Tr}(\eta(N, M)^T M^{-k} s s^T M^{-1}) - \cdots - \operatorname{Tr}(\eta(N, M)^T M^{-1} s s^T M^{-k}) .$$

The right-hand side of the above expression can be expressed as

$$- \sum_{i=0}^{k-1} \operatorname{Tr}(s^T M^{-(i+1)} \eta(N, M)^T M^{-(k-i)} s) .$$

By choosing $\eta(N, M) = M$, we get

$$s^T N^{-k} s - s^T M^{-k} s \geq - \sum_{i=0}^{k-1} \operatorname{Tr}(s^T M^{-k} s) ,$$

which implies

$$s^T N^{-k} s + \sum_{i=0}^{k-2} s^T M^{-k} s \geq 0 .$$

The above result follows because of the positive definiteness of $N$ and $M$. To complete the proof, we also need to show that if $\nabla f(M) = 0$, then $f(N) \geq f(M)$, $\forall N$, i.e., the stationary point is indeed the global minimum of the function. By equating the gradient to zero, we get

$$-M^{-k} s s^T M^{-1} = M^{-(k-1)} s s^T M^{-2} + \cdots + M^{-1} s s^T M^{-k} .$$

Right multiplication with $M$ gives us

$$-M^{-k} s s^T = M^{-(k-1)} s s^T M^{-1} + \cdots + M^{-1} s s^T M^{-(k-1)} ,$$

which implies

$$- \operatorname{Tr}(M^{-k} s s^T) = \operatorname{Tr}(M^{-(k-1)} s s^T M^{-1}) + \cdots + \operatorname{Tr}(M^{-1} s s^T M^{-(k-1)}) .$$

It follows that

$$- \operatorname{Tr}(s^T M^{-k} s) = \operatorname{Tr}(s^T M^{-k} s) + \cdots + \operatorname{Tr}(s^T M^{-k} s) ,$$

and thus

$$s^T M^{-k} s = 0 .$$

The above equation shows that this class of functions does not have any stationary point.

$\square$

### B.2 Proof for Theorem 4.4

*Proof.* Let $x = s^T K$. Then $f(L) = x^T(L+K)^{-1}K(L+K)^{-1}x$. The gradient of the function is given by

$$\nabla f(L) = -(L+K)^{-1}xx^T(L+K)^{-1}K(L+K)^{-1} - (L+K)^{-1}K(L+K)^{-1}xx^T(L+K)^{-1} .$$

Let $L_1, L_2 \in S_+^n$. To show invexity for function $f$, we need to show that there exists an $\eta(L_1, L_2)$ such that

$$f(L_1) - f(L_2) \geq \langle \eta(L_1, L_2), \nabla f(L_2) \rangle .$$

For our problem, this means that we need to show

$$\begin{aligned}
x^T(L_1+K)^{-1}&K(L_1+K)^{-1}x - x^T(L_2+K)^{-1}K(L_2+K)^{-1}x \geq \\
&- \langle \eta(L_1, L_2), (L_2+K)^{-1}xx^T(L_2+K)^{-1}K(L_2+K)^{-1} \rangle \\
&- \langle \eta(L_1, L_2), (L_2+K)^{-1}K(L_2+K)^{-1}xx^T(L_2+K)^{-1} \rangle \\
&= - \operatorname{Tr}(\eta(L_1, L_2)^T(L_2+K)^{-1}xx^T(L_2+K)^{-1}K(L_2+K)^{-1}) \\
&- \operatorname{Tr}(\eta(L_1, L_2)^T(L_2+K)^{-1}K(L_2+K)^{-1}xx^T(L_2+K)^{-1}) \\
&= - \operatorname{Tr}(x^T(L_2+K)^{-1}K(L_2+K)^{-1}\eta(L_1, L_2)^T(L_2+K)^{-1}x) \\
&- \operatorname{Tr}(x^T(L_2+K)^{-1}\eta(L_1, L_2)^T(L_2+K)^{-1}K(L_2+K)^{-1}x)
\end{aligned}$$

for a particular choice of $\eta(L_1, L_2)$. By choosing $\eta(L_1, L_2) = \frac{L_2+K}{2}$, we get

$$\begin{aligned}
x^T(L_1+K)^{-1}&K(L_1+K)^{-1}x - x^T(L_2+K)^{-1}K(L_2+K)^{-1}x \geq \\
&- \operatorname{Tr}(x^T(L_2+K)^{-1}K(L_2+K)^{-1}x) .
\end{aligned}$$

As $(L_1+K)^{-1}$ is a symmetric positive definite matrix, the matrix obtained by left multiplying it with a positive diagonal matrix is the same as right multiplying it with the same diagonal matrix and is positive definite. Thus

$$x^T(L_1+K)^{-1}K(L_1+K)^{-1}x = x^T(L_1+K)^{-1}K^{\frac{1}{2}}K^{\frac{1}{2}}(L_1+K)^{-1}x \geq 0 .$$

By following similar computation as shown in Theorem 4.2, it can be observed that the function has no stationary points and is $\eta$-invex for $\eta(\cdot, L) = \frac{(L+K)}{2}$. $\qquad\square$

### B.3 Proof for Theorem 4.5

*Proof.* From Theorem 4.2 we know that the class of functions $f(I+L) = s^T(I+L)^{-k}s$ are $\eta$-invex for $\eta(\cdot, L) = I + L$. Using the linearity of trace and partial derivative operators and following the similar computation as shown in Theorem (4.2), we can conclude that $\sum_{i=1}^{T} s^T(I+L)^{-2i}s$ is $\eta$-invex for $\eta(\cdot, L) = I + L$. $\qquad\square$

### B.4 Proof of Proposition 4.6

*Proof.* Recall that the Laplacian spectrum of the complete graph has an eigenvalue 0 with multiplicity 1 and an eigenvalue of $n$ with multiplicity $n-1$. When the opinions are mean-centered opinion vectors $s$ (such that $s^T 1 = 0$), the expressed opinions are given by $z = (I + L(K_n))^{-1}s = \frac{s}{n+1}$. The polarization of expressed opinions in the first time period is $z^T z = \|z\|^2 = \frac{\|s\|_2}{(n+1)^2}$. The $\mathcal{T}$-period polarization for the complete graph is

$$\frac{\|s\|_2}{(n+1)^2} + \frac{\|s\|_2}{(n+1)^4} + \cdots + \frac{\|s\|_2}{(n+1)^{2\mathcal{T}}} .$$

As each element in the above summation is the lower bound for the corresponding terms from the repeated polarization function, the global minimum for (9) is attained for $K_n$. $\qquad\square$

## B.5 Proof for Theorem 4.7

*Proof.* Consider the following two optimization problems:

$$
\begin{aligned}
&\underset{L \in \mathcal{L}}{\arg\min} \ \underset{s \in R^n, s \perp 1, \|s\|_2^2 \leq 1}{\max} && s^T(I+L)^{-2}s \\
&\text{subject to} && L_{ij} \in \{-1, 0\}, \text{ for } i \neq j \\
& && \| \operatorname{vec}(L) - \operatorname{vec}(L_0) \|_0 \leq 4k \ ,
\end{aligned}
\tag{17}
$$

and

$$
\begin{aligned}
&\underset{L \in \mathcal{L}}{\arg\max} && \lambda_2(L) \\
&\text{subject to} && L_{ij} \in \{-1, 0\}, \text{ for } i \neq j \\
& && \| \operatorname{vec}(L) - \operatorname{vec}(L_0) \|_0 \leq 4k \ .
\end{aligned}
\tag{18}
$$

Mosk-Aoyama (2008), showed that finding a set of edges within a specified budget to add to the graph so that the algebraic connectivity of the augmented graph is maximized is NP-hard. By Courant-Fischer theorem Golub & Van Loan (2013), we can observe that the inner maximization problem in (17) takes the maximum value of $\frac{1}{(1+\lambda_2(L))^2}$, when $s$, the mean-centered initial opinion vector, is the second smallest eigenvector of $L$. Thus for the outer minimization problem, we need an $L$ obtained from $L_0$ by adding $k$ edges and with maximum $\lambda_2$. The graph associated with the Laplacian matrix returned by equation (17) is the same as the solution of equation (18). Thus, the computational hardness of minimizing a restricted variant of polarization minimization (where $s$ is not a fixed vector) is at least that of maximizing algebraic connectivity within the budget $k$.

$\square$

## B.6 Proof of Theorem 5.1

*Proof.* In the following we represent $(I+L)^{-2}$ as $\begin{bmatrix} W_{11} & W_{12} \\ W_{12} & W_{22} \end{bmatrix}$, with each $W_{ij}$ being a block matrix having appropriate dimensions. For the sake of clarity, we omit the dimension details when they are evident from the context. For a given set of initial opinions vector $s = \begin{bmatrix} s_1^T & s_2^T \end{bmatrix}^T$, the polarization function can be expressed as follows:

$$
\begin{aligned}
f(L) = s^T(I+L)^{-2}s &= \begin{bmatrix} s_1^T & s_2^T \end{bmatrix} \begin{bmatrix} W_{11} & W_{12} \\ W_{12} & W_{22} \end{bmatrix} \begin{bmatrix} s_1 \\ s_2 \end{bmatrix} \\
&= s_1^T W_{11} s_1 + s_1^T W_{12} s_2 + s_2^T W_{12} s_1 + s_2^T W_{22} s_2
\end{aligned}
$$

On taking expectation with respect to the vector of unknowns $s_2$ we get

$$
\mathbb{E}(f(L)) = s_1^T W_{11} s_1 + \operatorname{Tr}(W_{22})
$$

Observe that the above equation can be rewritten as

$$
\mathbb{E}(f(L)) = a^T(I+L)^{-2}a + \sum_{i=1}^{m} b_i^T(I+L)^{-2}b_i
\tag{19}
$$

where $a = \begin{bmatrix} s_1^T & 0 \end{bmatrix}^T$ and $b_i = \begin{bmatrix} 0 & e_i^T \end{bmatrix}$ for all $i = \{1, \cdots, m\}$, with $e_i \in \mathbb{R}^m$ denoting the standard unit vector containing a 1 at its $i$-th entry. Notice that $a^T(I+L)^{-2}a$ and $\sum_{i=1}^{m} b_i^T(I+L)^{-2}b_i$ are $\eta$-invex. Using the linearity of trace and partial derivative operators and following the similar computation as shown in Theorem (4.2), we can conclude that $\mathbb{E}(f(L)) = a^T(I+L)^{-2}a + \sum_{i=1}^{m} b_i^T(I+L)^{-2}b_i$ is $\eta$-invex for $\eta(\cdot, L) = I + L$. $\square$

### B.7 Proof of Theorem 5.2

**Theorem B.1.** *Given a vector $s \in \mathbb{R}^n$ defined as $s = \begin{bmatrix} s_1^T & s_2^T \end{bmatrix}^T$, where $s_1 \in \mathbb{R}^{n-m}$ and $s_2 \in \mathbb{R}^m$, and assuming that $s_2$ is selected from a distribution satisfying $\mathbb{E}(s_2) = 0$ and $\mathbb{E}(s_2 s_2^T) = \Sigma$, it follows that $\mathbb{E}(f(L))$ is invex.*

*Proof.* Borrowing the notations from the Proof of Theorem 5.1, we represent $(I + L)^{-2}$ as $\begin{bmatrix} W_{11} & W_{12} \\ W_{12} & W_{22} \end{bmatrix}$, where each $W_{ij}$ is a block matrix with appropriate dimensions. For clarity, we omit dimension details when evident. For a given initial opinions vector $s = \begin{bmatrix} s_1^T & s_2^T \end{bmatrix}^T$, the polarization function is expressed as:

$$f(L) = s^T (I + L)^{-2} s = \begin{bmatrix} s_1^T & s_2^T \end{bmatrix} \begin{bmatrix} W_{11} & W_{12} \\ W_{12} & W_{22} \end{bmatrix} \begin{bmatrix} s_1 \\ s_2 \end{bmatrix}$$
$$= s_1^T W_{11} s_1 + s_1^T W_{12} s_2 + s_2^T W_{12} s_1 + s_2^T W_{22} s_2$$

Taking the expectation with respect to the vector of unknowns $s_2$, we obtain:

$$\mathbb{E}(f(L)) = s_1^T W_{11} s_1 + \mathbb{E}(s_2^T W_{22} s_2) \tag{20}$$
$$= s_1^T W_{11} s_1 + \mathbb{E}(\mathrm{Tr}(W_{22} s_2 s_2^T)) \tag{21}$$
$$= s_1^T W_{11} s_1 + \mathrm{Tr}(W_{22} \mathbb{E}(s_2 s_2^T)) \tag{22}$$
$$= s_1^T W_{11} s_1 + \mathrm{Tr}(W_{22} \Sigma), \tag{23}$$

where equation equation 22 follows due to the linearity of the trace function.

Since covariance matrix $\Sigma$ is a positive semidefinite matrix, it has a unique square root, i.e., $\Sigma = BB^T$ for a symmetric square matrix $B$. Using this property along with the cyclicity property of trace, we rewrite equation 23 as below:

$$\mathbb{E}(f(L)) = s_1^T W_{11} s_1 + \mathrm{Tr}(BW_{22}B) \tag{24}$$

If we represent $B = \begin{bmatrix} b_1 & b_2 & \cdots & b_m \end{bmatrix}$ for vectors $b_i \in \mathbb{R}^m, \forall i = \{1, \cdots, m\}$, then equation 24 can be expressed as:

$$\mathbb{E}(f(L)) = s_1^T W_{11} s_1 + \sum_{i=1}^{m} b_i^T W_{22} b_i, \tag{25}$$

which can be further rewritten as

$$\mathbb{E}(f(L)) = a^T (I + L)^{-2} a + \sum_{i=1}^{m} \bar{b}_i^T (I + L)^{-2} \bar{b}_i, \tag{26}$$

where $a = \begin{bmatrix} s_1^T & 0 \end{bmatrix}^T$ and $\bar{b}_i = \begin{bmatrix} 0 & b_i^T \end{bmatrix}^T$ for all $i = \{1, \cdots, m\}$. Recall that $a^T (I + L)^{-2} a$ and $\sum_{i=1}^{m} \bar{b}_i^T (I + L)^{-2} \bar{b}_i$ are $\eta$-invex. Using the linearity of trace and partial derivative operators and following the similar computation as shown in Theorem (4.2), we can conclude that $\mathbb{E}(f(L)) = a^T (I+L)^{-2} a + \sum_{i=1}^{m} \bar{b}_i^T (I+L)^{-2} \bar{b}_i$ is $\eta$-invex for $\eta(\cdot, L) = I + L$. $\qquad\square$

## C   Example to demonstrate the nonconvexity of the function $s^T M^{-2} s$

Here, we provide a visual depiction illustrating the nonconvex nature of the function $s^T M^{-2} s$, $M \in S_{++}^n$. In Figure 6, we plot the function $f(z) = s^T \begin{bmatrix} z & 0.9 \\ 0.9 & 1 \end{bmatrix}^{-2} s$ with respect to $z \in [1 \quad 2]$ and $s = \begin{bmatrix} 1 \\ 1 \end{bmatrix}$. Notice that this function is nonconvex.

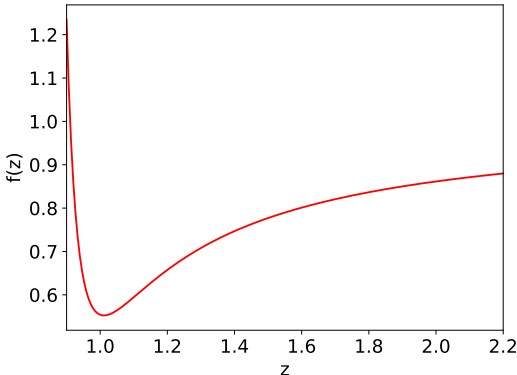

Figure 6: Nonconvexity of the function $s^T M^{-2} s$

## D   Additional Experiments

In this section we include further experimental results for fully known initial opinions.

**Karate Club network:**   This network represents a social conflict between an instructor and an administrator within a karate club, as documented by Zachary (1977). It is an undirected network comprising 34 nodes and 78 edges, where each node corresponds to a club member, and edges signify connections between members. Figure 1(a) illustrates the division of club members into two opinionated clusters due to the conflict. We attribute an initial opinion of "+1" or "-1" to each opinionated cluster.

In Figure 7, we present the polarization variations across different budget allocations for our invex relaxation model (optimization problem 13), the Coordinate Descent (CD) method, Tr minimization, and the Fiedler Difference (FD) approach. It is evident that the invex relaxation model consistently outperforms CD and other methods in terms of polarization reduction. FD reduces polarization by adding a single edge, resulting in the sparsest graph configuration. For our invex relaxation approach, by utilizing the thresholding parameter $|\rho| = 0.0002$, with 100 iterations of PGD, and employing a step size of $\alpha = 0.5$, the average number of edges across different budgets amounts to 184.

**Sawmill Strike network:**   This network represents employees working at a sawmill during a period of strike. It is an undirected network comprising 24 nodes and 76 edges. The strike's prolonged duration was believed to be due to ineffective communication between two distinct groups of employees within the network. The network was initially analyzed in Michael (1997) to identify leaders during the strike. In this study, we leverage this network to identify potential edges that could minimize polarization. We attribute an initial opinion of "+1" to one group and "-1" to another group of nodes.

Figure 7 (b) depicts the variation in polarization as the budget increases. Notably, our invex relaxation approach consistently achieves the most substantial reduction in polarization across different budget allocations when compared to the Coordinate Descent (CD) method. For our invex relaxation method, employing $|\rho| = 0.0002$ and with a step size of $\alpha = 0.5$, the average number of added edges amounts to 190.

**Preferential Attachment (Scale Free) Network:**   Preferential Attachment (PA) describes a mechanism of graph evolution where higher-degree nodes have a greater probability of receiving new neighbors. It is

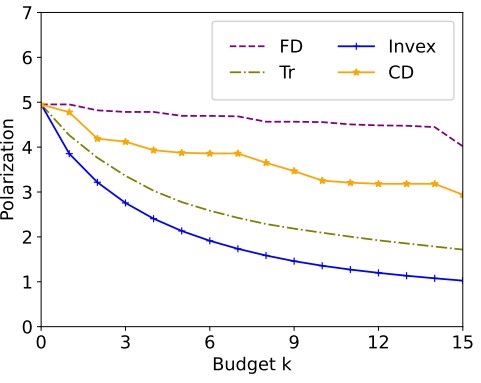 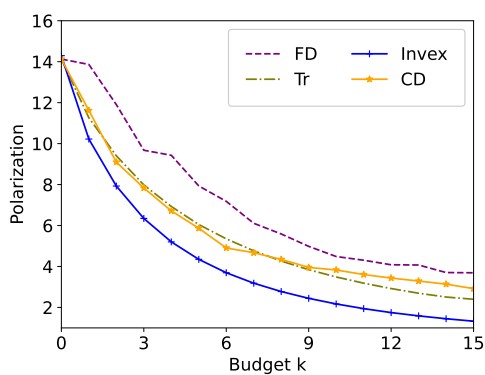

(a) Change in polarization with a budget on Karate Club network. Our nonconvex relaxation considerably reduces polarization compared to all other approaches.

(b) Change in Polarization on Sawmill Strike Network. Invex relaxation produces the best reduction in polarization compared to CD.

Figure 7: Polarization on Karate and Sawmill Networks

designed to model the power law behavior Faloutsos et al. (1999). For our analysis, an incoming vertex connects to at most four other existing vertices in the graph. The resultant PA network has 200 nodes and 768 edges. Nodes in the network are assigned an initial opinion of "+1" and "−1" uniformly at random.

Figure 8(a) visually illustrates the reduction in polarization across budgets ranging from $k = 1$ to $k = 15$. Notably, our invex relaxation method (optimization problem 13) consistently achieves the lowest polarization compared to other approaches. In our computational experiments, we executed projected gradient descent for up to 100 iterations, employing a step size of $\alpha = 0.8$ and $|\rho| = 0.0002$. On average, after applying the thresholding parameter $\rho$, the invex relaxation approach added $1,410$ edges across all budgets.

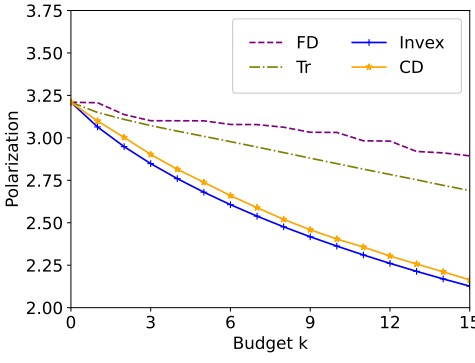 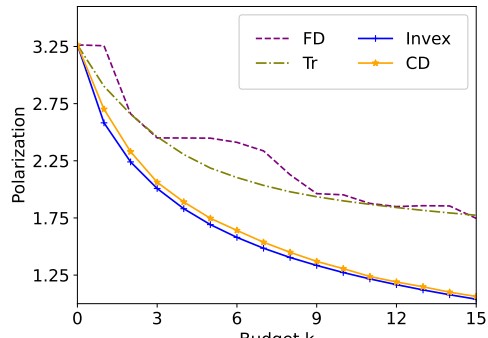

(a) Reduction in polarization with varying budgets using invex relaxation, CD, Tr, and FD approaches on Preferential Attachment network.

(b) Reduction in polarization with varying budgets using invex relaxation, CD, Tr, and FD approaches on Erdös-Rényi Graph.

Figure 8: Reduction in Polarization on Preferential Attachment and Erdös-Rényi graphs

**Erdös-Rényi:** In this model, each pair of vertices are connected independently with a probability $p$ Erdos & Renyi (1960). We construct an Erdös-Rényi graph with 100 vertices and $p = 0.1$. Nodes in the network are assigned an initial opinion of "+1" and "−1" uniformly at random. The step size for invex relaxation (optimization problem 13) is set to $\alpha = 0.8$. The projected gradient descent on 13 is run for 100 iterations with thresholding parameter $\rho = 0.0002$. The change in polarization is depicted in Figure 8(b).

# E Robustness of our approach towards small perturbations to Initial opinions

To assess the impact of inaccuracies in estimating the initial distribution, we conducted numerical experiments using a Stochastic Block Model (SBM) comprising 100 nodes and 1210 edges. Our focus lies only in the phase when initial opinions are unknown, with the objective of minimizing $\text{Tr}\langle \Sigma, (I + L)^{-2}\rangle$. We employed a perturbed (estimated) covariance matrix in experiments, denoted as $\hat{\Sigma}$. The construction of $\hat{\Sigma}$ involved slightly perturbing the eigenvalues of $\Sigma$. The subsequent paragraph presents the values for min $\text{Tr}\langle \hat{\Sigma}, (I+L)^{-2}\rangle$ and provide a comparison for minimizing the true value min $\text{Tr}\langle \Sigma, (I+L)^{-2}\rangle$. Perturbations were introduced to a subset of eigenvalues (in total, 40 eigenvalues are perturbed) by randomly adding or deleting values of $\delta$ (the range of the eigenvalues are $[0, 3)$). The polarization results for a specified budget ($k = 10$) are analyzed, with consistent trends observed across various budgets, ranging from $k = 1$ to 15. The initial polarization value without perturbation is 0.4117.

Table 1: Sensitivity to inaccuracies in estimating this initial distribution

| $\delta$ | POLARIZATION |
|---|---|
| 0.001 | 0.4121 |
| 0.01 | 0.4112 |
| 0.1 | 0.4209 |
| 0.25 | 0.3822 |
| 0.5 | 0.3243 |

As can be observed, the value of polarization remains closer to its true value for small perturbations.

