# OpenReview forum: "Minimizing Polarization from Partially to Fully Observable Initial Opinions"
_TMLR — Rejected by TMLR_

### Review · Reviewer_6TJy · 2025-02-05

**Summary Of Contributions:**

This paper studies a framework for minimizing the amount of polarization of opinions in the Friedkin Johnson model through adding/removing the edges in the social network. The main contribution of this paper is to show that the polarization function in Definition 2.1 is an invex function whose global optima is attained at every stationary point. Together with a standard $\ell_0$ to $\ell_1$ approximation heuristic, the paper proposes a projected gradient method to find a polarization-minimizing topology.

**Audience:**

Yes

**Broader Impact Concerns:**

None.

**Claims And Evidence:**

No

**Requested Changes:**

See the above points on weaknesses.

**Strengths And Weaknesses:**

The major strength of the paper is that it offers a new perspective for the polarization minimization problem through directly minimizing the polarization cost. (P.S. provided its claim on novelty is valid, which the reviewer is unable to verify given the short review period of TMLR; also see the following discussions on weaknesses.)

However, the paper also has a number of weaknesses which outweigh its strength and has led to doubts over its technical correctness, as listed in order:

*The writing quality is poor*. Besides the lack of motivation as to why one would seek to modify the graph topology to minimize polarization, a large portion of the technical definitions & results were stated without any justification. For example, what's the point for introducing Definition 2.1-2.3? why do these definitions make sense? what're the main takeaways from Lemma 4.4 to 4.6, why are they important?

Moreoer, regarding the contributions over prior works, it is unclear what are main differences between the proposed approach with them. In particular, how did the prior works such as Musco et al., 2018, Xu et al., 2018 get around with the non-convexity of the polarization cost in Definition 2.1.

Notice that it has significantly affected the readability of the paper about the correctness of its contribution(s).

(P.S. It is understood that Definition 2.1 was proposed by older papers such as Musco et al. 2018, yet it is not appropriate for the current paper to skip all the context.)

*Lack of technical rigor*. In addition to the lack of justifications for a number of technical claims, the paper claimed that the polarization objective is invex and therefore can be solved by finding a stationary point of it. However, the original formulation contains an $\ell_0$ constraint on the topology perturbation which is then approximated into that of an $\ell_1$ constraint. Note that this approximation is at best a heuristic unless further theories can be established for the solution quality. Unfortunately this is a crucial step which may undermine the invex objective function property that has been established in the paper.

P.S. Strictly speaking (9) is not a "relaxation" in the context of the current paper since the feasible set in (9) is not necessarily a superset for that of (6).

In Sec 5, why can one assume that the unobserved initial opinion is Gaussian?

---

### Review · Reviewer_CJCm · 2025-02-22

**Summary Of Contributions:**

## Summary

This paper studies the problem of minimizing polarization within a network based on the Friedkin-Johnson (FJ) model.
They show that for this problem, the objective function is invex, indicating that every local minimum is a global minimum, despite being non-convex.
They propose a continuous relaxation to minimize the polarization function:
$ f(L) = s^{\top} (I+L)^{-2}s $
with edit constraints and a budget   $k $, considering both scenarios where initial opinions are partially or fully known.
They demonstrate the proposed methods in empirical experiments conducted in real-world network scenarios.
All results rely on the assumption that the unknown opinion  $s_2$ is generated such that:
$ E[s_2] = 0, \quad E[s_2 s_2^{\top}] = I $,
thereby satisfying the invexity condition.

**Audience:**

Yes

**Claims And Evidence:**

No

**Requested Changes:**

Please see the Weakness sections for improvements. Additionally following points could be addressed in the revised version.


## Related Work

The following paper can be cited in the related work section as it addresses the case where initial opinions are unknown in the FJ model:

Sublinear-Time Opinion Estimation in the Friedkin--Johnsen Model.
  Stefan Neumann, Yinhao Dong, Pan Peng, 2024 ACM Web Conference.

## Minor Concerns

- The use of Lemma, Theorem, and Corollary could be clearer. If a statement is important, it should be labeled as a Proposition, Theorem, or Corollary rather than repeatedly referring to Lemmas.
- There are many typos, such as “equation equation X" in various places. Use `\eqref{}` or cleverref to standardize the formatting.
- Citation formatting contains typos. In English writing, references should use (Authors, Year) style unless they serve as the subject or object of a sentence.
- The structure of Section 4  is strange; Section 4.1 appears after a list of results, making the organization unclear.
- When referring to optimization problem (8), it should be called an optimization problem rather than just equation 8.
- The conclusion should be a standalone section rather than placed at the end of Section 6.2.



Question (minor)

- In the continuous relaxation, my understanding is that arbitrarily small weights (e.g., adding edges with weight  0.00001) are allowed. If so, it seems that the global minimum would be the closest complete graph to  $L_0$. The experiments use a threshold parameter to remove edges with small weights—what is the intention behind this choice?

**Strengths And Weaknesses:**

## Strengths

- The problem setting, where only a subset of users' initial opinions is publicly accessible, is very important to investigate given real-world applications.

## Weaknesses

The writing is very confusing.

- Definitions 2.1, 2.2, 2.3, and 2.5 define the objective function  $P(z)$, but Definition 2.4 is presented as an optimization problem.
- Problem 1 and Problem 2 only describe research questions and are not mathematically well-defined as optimization problems. The distinction between fully known and partially known initial opinions is mentioned, but a rigorous optimization problem definition is missing.
- In Section 4, it is unclear what the objective function is and which minimization problem is being referred to. In particular, the NP-hardness claim in Lemma 4.6 is ambiguous—what exactly is the optimization problem being discussed? What is the objective function, what are the constraints, and what constitutes a feasible solution? The term "adversarial polarization" is also undefined. When given a budget $ k$, what types of interventions are allowed? In the discrete setting, are only edge deletions and additions permitted (i.e., unit weight interventions)?
- The writing style also has issues; $ L_0 $ should be clearly defined as an input in the optimization problem.
- Section 5 and other sections suffer from the same lack of rigorous definitions.

Significance of Methodology

- Section 5 deals with the objective function:

  $$ f(L) = s^{\top} (I+L)^{-2}s $$

  and introduces an additional assumption about the unknown opinion generation:

  $$ E[s_2] = 0, \quad E[s_2 s_2^{\top}] = I. $$

  Theorem 5.1 claims invexity under this assumption. However, the real-world implications and validity of this assumption should be discussed. Additionally, if this assumption is crucial, it should be explicitly stated in the introduction rather than implied indirectly.
- With this assumption, the objective function can be rewritten using a block matrix decomposition of $ (I+L)^{-2} $ as:

  $$ f(L) = s_1^{\top} W_{11} s_1 + \text{Tr}(W_{22}) $$

  leading to optimization problem (8), but this derivation is quite straightforward.
- Section 6 presents experiments on Problems 1 and 2, but only for the relaxed optimization problem defined in Definition 2.4 (and Definition 5). Why are Definitions 2.1–2.3 not addressed? Are stubborn actors or the multi-period setting introduced in Section 4.1 not considered in the experiments?

3. Significance of Experiments

- The performance difference between the proposed method and Coordinate Descent (CD) seems incremental.
- Scalability can be more addressed. The sizes of the networks used in the experiments should be specified. Additionally, reporting the computational time required would improve the analysis.
- The experiments use $ s_2 = \text{mean}(s_1) $ in Coordinate Descent, but since $ E[s_2] = 0 $ isis assumed, what happens when $ s_2 = 0 $ instead?

---

### Review · Reviewer_1Ayj · 2025-02-23

**Summary Of Contributions:**

This paper considers the problem of optimizing polarization in networks based on the Friedkin-Johnson (FJ) model of opinion dynamics.

The authors first provide a review of prior works along this line of research, and then analyze basic properties such as invexity analysis of certain functions related to graph Laplacian. Moreover, they give a nonconvex relaxation for optimization of polarization by replacing the l0-norm constraint (typically considered in the standard setting) with an l1-norm constraint. Finally, numerical simulations are given to further demonstrate the effectiveness of their methods.

**Audience:**

Yes

**Broader Impact Concerns:**

Although the authors make contributions in theoretically analyzing the problem and empirically conducting numerical experiments, I feel that the topics discussed may not be closely related to machine learning community. Instead, a SIAM journal on optimization, graph theory or computational mathematics might be a better fit for this paper. Or the authors may want to highlight the applications of this study to machine learning problems.

**Claims And Evidence:**

Yes

**Requested Changes:**

1. (Technical novelty) Given that the major part of the paper is dedicated to solving polarization optimization, it is worth analyzing the connections between the solution of relaxed problem and the solution of the original problem.

2. (Machine learning relevance) The paper mainly considers the optimization problems on graphs. The connections between the proposed methods and machine learning seems unclear to the readers. The authors may want to highlight the relevance of the methods to the machine learning community.

**Strengths And Weaknesses:**

Strengths:

1. The paper is well-written and easy to follow.
2. Theoretical results are sound and empirical experiments are reasonable to validate the theory.

Weaknesses:

1. The major weakness is the technical novelty of nonconvex relaxation of polarization optimizatio. This type of relaxation has been widely used in optimization literature.

2. The theoretical results are only limited to basic properties of some functions such as invexity, and there seems no theorems on analyzing the properties of the solutions of the relaxed problem.

---

### Decision · Action_Editor_WS85 · 2025-04-18

**Recommendation:** Reject

**Comment:**

This paper considers the problem of optimizing polarization in networks based on the Friedkin-Johnson model of opinion dynamics. Its contributions include analyzing the convexity of certain functions related to graph Laplacian, providing a nonconvex relaxation for optimizing polarization by an l1-norm constraint, and numerical simulations.

Two reviewers raised concerns about multiple issues in the paper related to mathematical notation, motivation, equation references, and presentation quality. The concerns prevailed after the rebuttal round. In the present, it is difficult to assess the technical correctness with confidence.

Hence, I reject the paper in its current form. However, I leave open the possibility of re-submitting it at a later time after it undergoes a  major revision carefully addressing all concerns from the reviews.

I suggest extending the motivation and experimental evaluation sections to further validate the paper's appropriateness for the machine learning community.

**Audience:**

I think that there is a non-trivial TMLR audience  interested in the paper findings, given that  several related work were published in the Neurips conference proceedings.

**Claims And Evidence:**

This paper considers the problem of optimizing polarization in networks based on the Friedkin-Johnson (FJ) model of opinion dynamics. The paper's contributions include: analysis of invexity of certain functions related to graph Laplacian; provides a nonconvex relaxation for optimization of polarization by an l1-norm constraint; and numerical simulations are performed. Two reviewers raised concerns about multiple issues in the paper related to mathematical notation, the motivation, equation references, and presentation quality.

**Resubmission Of Major Revision:**

The authors may consider submitting a major revision at a later time.